# Combining tubercidin and cordycepin scaffolds results in highly active candidates to treat late-stage sleeping sickness

Fabian Hulpia [1], Dorien Mabille [2], Gustavo D. Campagnaro[3], Gabriela Schumann[4], Louis Maes[2], Isabel Roditi [4], Anders Hofer[5], Harry P. de Koning [3,6], Guy Caljon [2,6] & Serge Van Calenbergh [1,6]*

African trypanosomiasis is a disease caused by *Trypanosoma brucei* parasites with limited treatment options. *Trypanosoma* is unable to synthesize purines de novo and relies solely on their uptake and interconversion from the host, constituting purine nucleoside analogues a potential source of antitrypanosomal agents. Here we combine structural elements from known trypanocidal nucleoside analogues to develop a series of 3'-deoxy-7-deazaadenosine nucleosides, and investigate their effects against African trypanosomes. 3'-Deoxytubercidin is a highly potent trypanocide in vitro and displays curative activity in animal models of acute and CNS-stage disease, even at low doses and oral administration. Whole-genome RNAi screening reveals that the P2 nucleoside transporter and adenosine kinase are involved in the uptake and activation, respectively, of this analogue. This is confirmed by P1 and P2 transporter assays and nucleotide pool analysis. 3'-Deoxytubercidin is a promising lead to treat late-stage sleeping sickness.

--------------------------------------------------------------------------------

[1] Laboratory for Medicinal Chemistry (Campus Heymans), Ghent University, Ottergemsesteenweg 460, 9000 Gent, Belgium. [2] Laboratory of Microbiology, Parasitology and Hygiene (LMPH), University of Antwerp, Universiteitsplein 1, 2610 Wilrijk, Belgium. [3] College of Medical, Veterinary and Life Sciences, Institute of Infection, Immunity and Inflammation, University of Glasgow, Glasgow G12 8TA, UK. [4] Institute of Cell Biology, University of Bern, Baltzerstrasse 4, 3012 Bern, Switzerland. [5] Department of Medical Biochemistry and Biophysics, Umeå University, 901 87 Umeå, Sweden. [6]These authors contributed equally: Harry P. de Koning, Guy Caljon, Serge Van Calenbergh *email: Serge.VanCalenbergh@UGent.be

**S**leeping sickness or human African trypanosomiasis (HAT) is almost always fatal and is endemic in much of sub-Saharan Africa, coinciding with the geographical localization of the tsetse fly vector. Its causative agent is the haemoflagellate protozoan parasite *Trypanosoma brucei* spp., of which *T. b. gambiense* and *T. b. rhodesiense* are infectious to humans, and prevalent in West and Central Africa, and in East and Southern Africa, respectively[1]. Patients initially show non-specific symptoms such as fever and general malaise, caused by parasites proliferating in the haemolymphatic system (stage 1 disease), after which the trypanosomes invade the central nervous system (CNS; stage 2 disease), thereby causing severe neurological complications, one of which is the altered sleep/wake cycle that gave this infectious disease its name[2–4].

Treatment of HAT is currently based on the following five drugs: pentamidine, suramin, melarsoprol, eflornithine and nifurtimox[5]. A sixth drug, fexinidazole, recently concluded clinical trials successfully[6]. Pentamidine and suramin are the first-line drugs against stage 1 disease caused by *T. b. gambiense* and *T. b. rhodesiense*, respectively. The first-line treatment for the second stage of *T. b. gambiense* HAT is a nifurtimox–eflornithine combination therapy, with eflornithine monotherapy used when nifurtimox is unavailable or contraindicated. Melarsoprol, an organo-arsenical compound, leads to treatment-related death in 2.5 to 5% of cases[7,8] and is now restricted to the treatment of stage 2 *T. b. rhodesiense* HAT, while being almost completely phased out for stage 2 *T. b. gambiense* HAT. All these drugs suffer from major limitations ranging from stage-specific efficacy (e.g. only active against stage 1 disease) to significant toxicity, as well as the necessity for parenteral administration (intravenous for suramin, melarsoprol and eflornithine and intramuscular for pentamidine), which poses practical challenges in rural Africa. Clinical trial results with orally administered fexinidazole[6] showed it is safe and effective against *T. b. gambiense* HAT, marking it the first new HAT therapeutic in three decades, as well as the first oral monotherapy against both stage 1 and stage 2 HAT. Nonetheless, resistance is readily induced in vitro and fexinidazole displays cross-resistance with nifurtimox[9,10]. Additionally, this drug requires a high pill burden treatment regime[6], underscoring that research efforts for the discovery of new therapeutics to treat this neglected tropical disease remain of significant interest[2,3].

Protozoan parasites are incapable of synthesizing purine nucleosides de novo and hence rely on uptake and salvage of exogenous purines. In this context, purine analogues that can act as inhibitors[11–13] or 'subversive' substrates[14] of purine salvage enzymes are a promising source of compounds with activity against protozoan parasites (e.g. cordycepin[15–18], formycin B[16] and tubercidin[19,20]) and have been shown to exhibit good activity against African trypanosomes[14,17,18,21]. Moreover, nucleoside analogues could have the advantage of a higher likelihood to cross the blood–brain barrier (BBB) and thus be active against stage 2

HAT, owing to the presence of specific (purine) transporters at the BBB[22]. The nucleoside antibiotics cordycepin 3[15–17,23] and tubercidin 6[19,24] represent two of the most thoroughly studied antitrypanosomal nucleoside analogues (Fig. 1).

Inspired by the activity of tubercidin against *T. brucei* spp., we recently explored a series of 7-substituted tubercidin analogues and identified analogues displaying promising in vitro activity against kinetoplastid parasites[25]. In an attempt to further increase the antitrypanosomal activity, we set out to investigate the effect of modifying the sugar part of tubercidin and its 7-substituted analogues.

The present communication reports the identification of a promising adenosine analogue that is highly active in both stage 1 and stage 2 mouse models of HAT. Furthermore, we demonstrate its affinity for *T. brucei* adenosine transporters, and provide insights into its mechanism of action by applying whole-genome RNA interference (RNAi) screening, and analysis of its metabolism in the parasite through nucleotide pool analysis.

## Results

**Hybrid nucleosides display highly potent in vitro activity.** Based on the reported activity of cordycepin and tubercidin, and taking into account our recently reported tubercidin derivatives[25], a small series of 2′- and 3′-deoxytubercidin analogues was synthesized (Supplementary Methods) and evaluated in vitro against *T. brucei* spp. (Fig. 2).

Comparison of the in vitro activity profile of the ribofuranose (6 and 8), 2′-deoxy (11 and 12) and 3′-deoxyribofuranose analogues (9 and 10) showed a clear preference for the latter with respect to antitrypanosomal potency, as well as selectivity (Table 1).

Moreover, we observed that the effect of most of the analogues remained almost unchanged (<3-fold) when assayed against drug-resistant *T. brucei* strains (Table 2), except for the 11-fold reduction in sensitivity to 9 observed in trypanosomes lacking the *TbAT1* gene encoding the P2 aminopurine transporter, that is, TbAT1-knockout (KO) and clone B48 (derived from TbAT1-KO, with additional loss of the HAPT1/AQP2 transporter)[26]. Importantly, the absence of the P2 transporter did not cause insensitivity to 9 as its in vitro activity remained submicromolar. It is therefore unlikely that the P2 transporter is the only transport protein involved in the uptake of 9, although it is probably the most important one. The resistance pattern shows no indication that any of the nucleosides are substrates of the HAPT1/AQP2 drug transporter, given the similar effect observed on B48, which is deficient in this activity, and its parental cell line, TbAT1-KO. Surprisingly, several analogues (6, 8 and 12) displayed reduced activity to the isometamidium-resistant cell line ISMR1, which is not known to have altered nucleoside transport. This may however be related to the fact that these parasites lack kinetoplasts and have reduced mitochondrial membrane potential, resulting in a substantially slower growth rate[27].

**Fig. 1** Different nucleoside analogues with reported activity against African trypanosomes. [Cordycepin: TCMDC-143080; Formycin B: TCMDC-143083 (codes originating from ref. [16])].

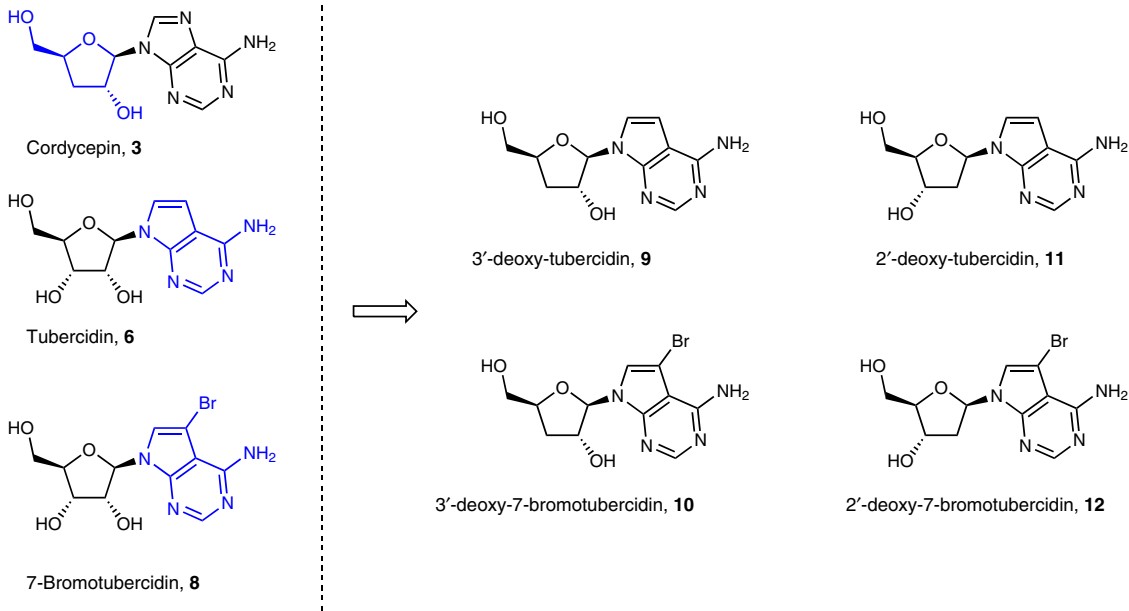

**Fig. 2** Combination of cordycepin and tubercidin yields highly potent hybrids (Table 1).

### Table 1 In vitro drug sensitivity of 7-deazapurine nucleoside analogues.

| Cpd. | *T. b. brucei* EC$_{50}$ (µM) | *T. b. rhodesiense* EC$_{50}$ (µM) | MRC-5 EC$_{50}$ (µM) | SI (MRC/*T.b. brucei*) | SI (MRC/*T.b. rhodesiense*) |
|---|---|---|---|---|---|
| Tubercidin, **6** | 0.48 ± 0.1 | 0.036 ± 0.001 | 2.2 ± 0.7 | 4 | 61 |
| **8** | 1.2 ± 0.3 | 0.12 ± 0.02 | 12 ± 2 | 10 | 107 |
| **9** | 0.048 ± 0.009 | 0.00052 ± 0.00004 | >64 | >1333 | >123,077 |
| **10** | 0.0013 ± 0.0003 | 0.00040 ± 0.00009 | 15 ± 3 | 11,462 | 37,500 |
| **11** | 48 ± 1 | >64 | >64 | 1.3 | – |
| **12** | 0.46 ± 0.08 | 0.12 ± 0.01 | 6.1 ± 0.7 | 13 | 50 |

EC$_{50}$ values, determined with the Alamar blue cell viability assay, are expressed in µM, and represent mean and SEM. The number of independent replicates was $n = 5$ (**8**, **9**, **10**), $n = 4$ (**6**) and $n = 2$ (**11** and **12**). Cytotoxicity was assessed in human MRC-5 fibroblasts. Source data are provided as a Source Data file
*SI* selectivity index

### Table 2 In vitro drug sensitivity against drug-resistant *T. brucei*.

| Cpd. | Lister 427 EC$_{50}$ (µM) | TbAT1-KO EC$_{50}$ (µM) | RF | B48 EC$_{50}$ (µM) | RF | ISMR1 EC$_{50}$ (µM) | RF |
|---|---|---|---|---|---|---|---|
| **6** TUB | 0.15 ± 0.03 | 2.61 ± 0.70 | 17.2** | 4.3 ± 1.3 | 28.7* | 1.7 ± 0.5 | 11.1** |
| **8** 7-Br-TUB | 0.32 ± 0.27 | 0.825 ± 0.0173 | 2.62*** | 0.448 ± 0.196 | 1.42 | 1.529 ± 0.157 | 4.85** |
| **9** 3′-Deoxy-TUB | 0.033 ± 0.001 | 0.375 ± 0.012 | 11.2*** | 0.368 ± 0.013 | 11.0*** | 0.0324 ± 0.0038 | 0.97 |
| **10** 7-Br-3′-deoxy-TUB | 0.0018 ± 0.0003 | 0.0021 ± 0.00025 | 1.2 | 0.0013 ± 0.0003 | 0.75 | 0.0011 ± 0.0002 | 0.63 |
| **11** 2′-Deoxy-TUB | 96.4 ± 13.8 | 43.7 ± 2.2 | 0.45** | 41.8 ± 1.4 | 0.43** | 77.8 ± 1.7 | 0.81 |
| **12** 7-Br-2′-deoxy-TUB | 0.31 ± 0.063 | 0.76 ± 0.18 | 2.37* | 0.83 ± 0.16 | 2.64** | 0.92 ± 0.12 | 2.93** |
| Pentamidine | 0.011 ± 0.001 | 0.018 ± 0.002 | 1.8** | 0.99 ± 0.16 | 94.6*** | 0.14 ± 0.04 | 13.8** |
| Diminazene | 0.42 ± 0.064 | 4.5 ± 0.9 | 10.6*** | 7.2 ± 1.6 | 16.9*** | 2.9 ± 0.4 | 6.9*** |
| Isometamidium | 0.65 ± 0.085 | 0.75 ± 0.14 | 1.2 | 0.56 ± 0.13 | 0.85 | 3.1 ± 0.5 | 4.8*** |

In vitro antitrypanosomal evaluation against drug-resistant *T. b. brucei* cell lines. EC$_{50}$ values are given in µM and are mean and SEM of three independent determinations ($n = 3$). RF = Resistance factor, that is, the ratio of EC$_{50}$ of resistant and reference (Lister 427WT) cell line. TbAT1-KO: *T. brucei* cell line lacking the *TbAT1*(P2) transporter gene. B48: pentamidine, diminazene and melaminophenyl arsenical resistant *T. brucei* line. ISMR1: isometamidium-resistant *T. brucei* cell line. TUB, tubercidin. Source data are provided as a Source Data file
*$p < 0.05$, **$p < 0.01$; ***$p < 0.001$; Student's $t$ test (two-sided)

**3′-Deoxytubercidin analogues are trypanocidal**. Incubation of wild-type *T. brucei* cells with **8**, **9** and **10** at 2× and 5× their half-maximal effective concentration (EC$_{50}$) showed that derivatives **9** and **10** display a clear trypanocidal effect against *T. brucei* bloodstream forms in vitro, whereas cultures incubated with **8** at either 2× and 5× EC$_{50}$ only displayed growth arrest after a few hours, demonstrating a more trypanostatic characteristic (Supplementary Fig. 1 and Supplementary Fig. 2). We further observed that cells treated with adenosine analogues **8**, **9** and **10** at 2× and 5× EC$_{50}$ showed clear aberrant morphology at 6 h and especially at 12 h, with gross distortions of cell shape and the apparent formation of intracellular vacuoles. Staining with the fluorescent DNA dye DAPI (4′,6-diamidino-2-phenylindole) revealed no evidence of inhibition of cytokinesis, which usually

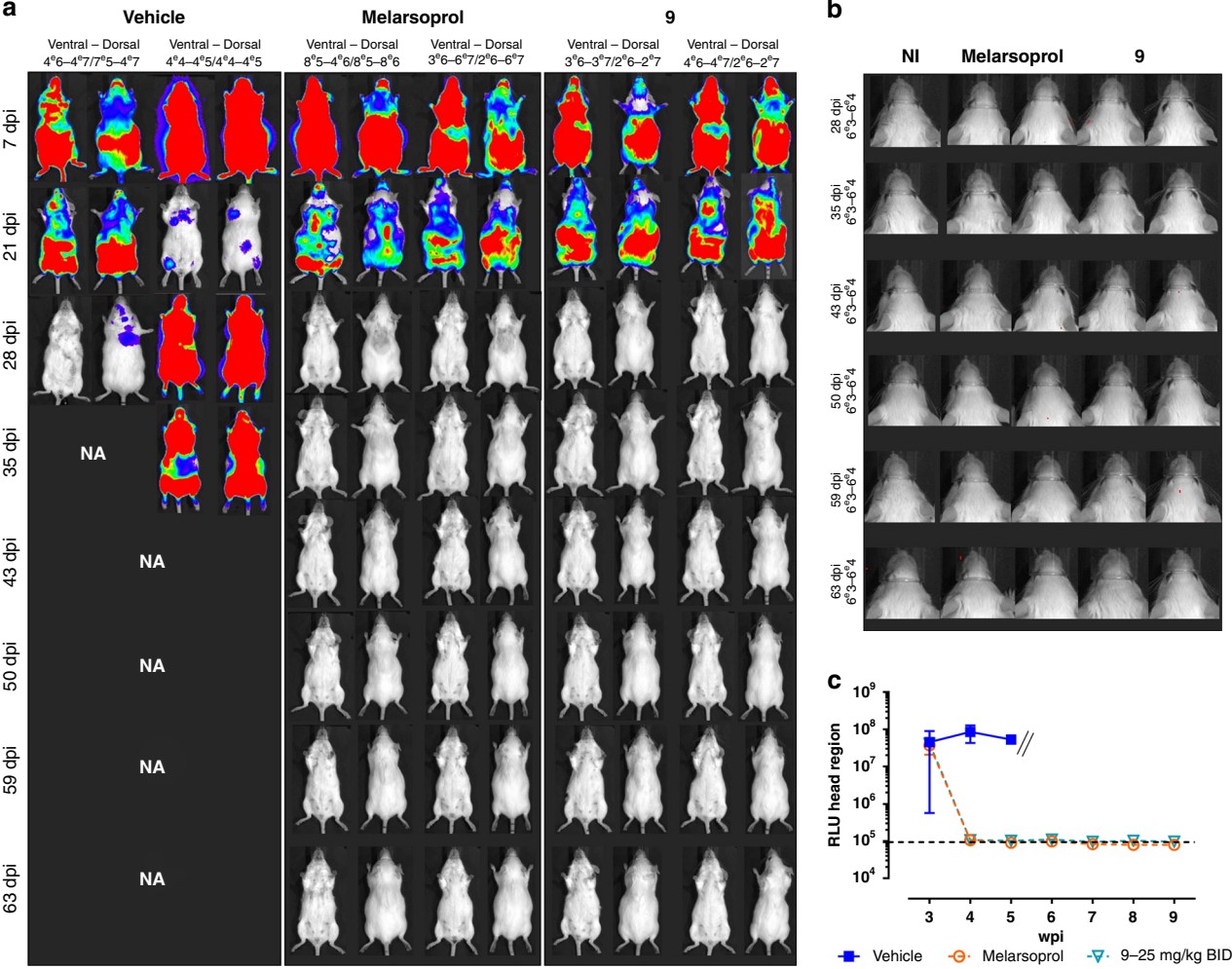

**Fig. 3** In vivo evaluation of **9** in a stage 2 mouse model of HAT. **a** BLI images of stage 2 *T. b. brucei*-infected mice treated at 21 dpi with vehicle ($n = 2$), melarsoprol ($n = 2$) or **9** ($n = 2$). Images were acquired from the dorsal and ventral sides of the animal. Image scales are indicated for each animal. NA = data not available due to animals in the control group succumbing to infection. **b** BLI images of the head region of melarsoprol and compound **9**-treated mice, with a non-infected animal (NI) as a reference. **c** BLI (average signal ± SEM) of the head region. // = data not available due to animals in the control group succumbing to infection. Dashed lined represents the background signal in a non-infected animal. Source data are provided as a Source Data file.

leads to the appearance of large, multi-nucleated cells (Supplementary Fig. 3). The morphology of the nucleus and kinetoplast appears not to have been substantially altered in compound-treated cell populations.

**Nucleoside analogue 9 is metabolically stable.** Incubations with mouse, rat and human S9 microsomal fractions revealed that **9** is metabolically stable, defined as ≥50% of parent compound remaining after 30 min (Supplementary Table 1). Next, the impact of adenosine deaminase on 7-deazapurine nucleoside analogues **6**, **9** and **10** was studied, given that it has been described to greatly affect the antitrypanosomal activity of adenosine analogues[17,18,23]. Therefore, the in vitro antitrypanosomal activity was re-assessed in the presence of an inhibitor of this enzyme, 2′-deoxycoformycin (Supplementary Table 2). In contrast to cordycepin (**3**), the $EC_{50}$ values of the 7-deazapurine analogues (**6**, **9** and **10**) were not significantly affected by the inhibition of adenosine deaminase.

Based on its low cytotoxicity, potent in vitro activity (Fig. 2) and metabolic stability, nucleoside analogue **9** was selected for follow-up evaluation in mouse models of HAT.

**Compound 9 shows excellent efficacy in murine HAT models.** First, **9** was evaluated in a mouse model of acute HAT. Intraperitoneal administration at 10 mg kg⁻¹ s.i.d. (once a day) for 5 days, or oral dosing at 25, 12.5 and 6.25 mg kg⁻¹ b.i.d. (twice a day) for 5 days resulted in negative blood parasitaemia and survival of all treated mice at the pre-set endpoint of 21 days post infection (dpi) (Supplementary Fig. 4). No adverse reactions have been noted nor was weight loss recorded in any of the treatment schedules, indicating that **9** was well tolerated by the test animals. Unfortunately, the bromo analogue **10**, although more potent in vitro, was not well tolerated at a dose of 6.25 mg kg⁻¹ orally, and was not pursued further.

Next, **9** was evaluated in a CNS mouse model of HAT. Oral dosing at 25 mg kg⁻¹ b.i.d. for 5 days resulted in excellent activity, comparable to the reference drug for CNS clearance, melarsoprol. All treated animals showed negative blood parasitaemia and survived the pre-set endpoint of 9 weeks post infection (wpi). Bioluminescent imaging (BLI) confirmed a total clearance of all organs, including the CNS (Fig. 3a, b). Quantification of the luminescent signal in the head region showed a rapid decline of the signal in both melarsoprol and **9**-treated animals to levels similar to those detected in a non-infected control mouse

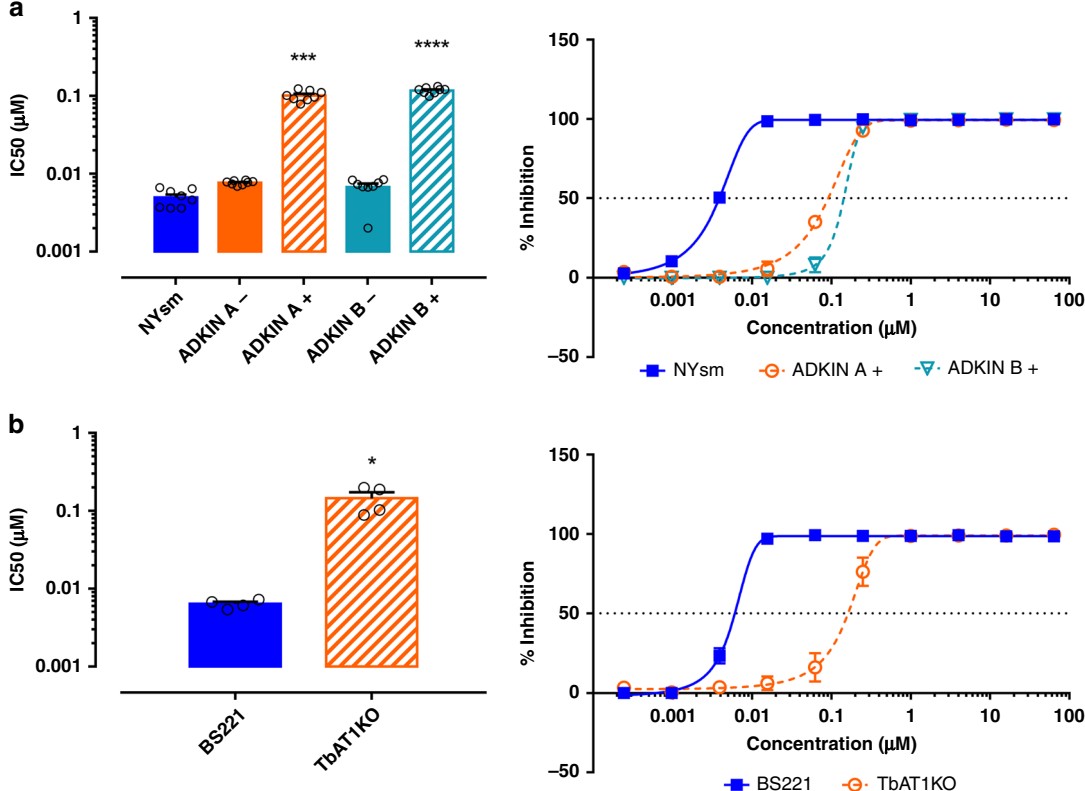

**Fig. 4** Susceptibility of ADKIN RNAi-mediated knockdown and TbAT1-KO cell lines to **9**. **a** Sensitivity of the RNAi clones targeting adenosine kinase (ADKIN). **b** Sensitivity of AT1-KO *T. b. brucei* strain, compared to the BS221 cell line from which it was derived. Results are expressed as the mean $IC_{50}$ (μM), and error bars represent SEM and are based on two independent experiments ($n = 2$), each with two biological replicates. $+ =$ tetracycline-induced clones; $- =$ non-induced clones. All experiments were performed with two independently generated RNAi clones (**a**, **b**). *$P < 0.05$, ***$p < 0.001$, ****$p < 0.0001$, Kruskal–Wallis test with Dunn's multiple comparison test. Source data are provided as a Source Data file.

(Fig. 3c). Dose titration at 25, 12.5 and 6.25 mg kg$^{-1}$ s.i.d. for 5 days also resulted in negative blood parasitaemia up to 9 wpi, with BLI confirming a total clearance in all organs including the CNS (Supplementary Fig. 5). Additional follow-up of surviving mice with BLI up to 14 wpi did not reveal any signs of relapse. All treated (melarsoprol and analogue **9**) animals showed complete absence of parasite burdens as assayed in several tissue samples (brain, spleen and fat) by a highly sensitive spliced-leader RNA (SL-RNA) quantitative PCR (qPCR) detection method (Supplementary Table 3)[28].

**Whole-genome RNAi screening**. Exposure of a genome-wide *T. b. brucei* RNAi library to **9** led to the selection and identification of six RNAi inserts that reduced the in vitro effectiveness of **9** (Supplementary Table 4). In order to confirm the involvement of the different inserts in the mode of action of **9**, the RNAi constructs were back-cloned into *T. brucei* NY-SM cells, and for each, two independent clones containing genomically inserted RNAi cassettes were tested for their susceptibility to the compound (except for the *AT1* gene, as a KO (AT1-KO) is available).

Out of the five knockdown constructs and the AT1-KO tested, only the knockdown targeting adenosine kinase (*ADKIN*; Tb927.6.2300) and the adenosine transporter 1 (*AT1*; Tb927.5.286b) KO cell line decreased the sensitivity for **9**: from an $EC_{50}$ of $0.0077 \pm 0.00017$ μM (mean and SEM, $n = 2$) to $0.10 \pm 0.005$ μM (mean and SEM, $n = 2$) upon ADKIN knockdown (Fig. 4a) and from $0.0064 \pm 0.0004$ μM (mean and SEM, $n = 2$) to $0.15 \pm 0.03$ μM (mean and SEM, $n = 2$) upon AT1-KO (Fig. 4b). Notably, despite the ~20-fold reduced susceptibility observed in the ADKIN knockdown cell lines, the in vitro activity of **9**

remained in the submicromolar range, suggesting that both *ADKIN* and *AT1* contribute to an important extent but are not essential per se for the antitrypanosomal activity of **9**. However, it must also be realized that the RNAi knockdown of *ADKIN* is unlikely to be 100% and that residual ADKIN activity therefore may be responsible for the remaining activity. For the remaining four constructs, no significant differences in susceptibility could be observed between tetracycline-induced and non-induced clones (Supplementary Fig. 6).

**Adenosine transporter assays**. The polar nature of the presented nucleoside analogues renders transmembrane translocation through passive diffusion highly unlikely. To confirm the role for the TbAT1/P2 aminopurine transporter in the drug sensitivity of trypanosomes to analogue **9** and to assess the contribution of the P1 purine nucleoside transporter[29] in the uptake of this and structurally related nucleosides, we investigated analogues **8**, **9** and **10** for their ability to inhibit P1- and P2-mediated [$^3$H] adenosine uptake (Fig. 5). The $K_m$ values for adenosine, obtained as control, were consistent with previously reported values (Table 3)[29,30].

Regarding transport via P1, we observed that all three analogues, and especially **9**, presented substantially lower affinity than adenosine (Fig. 5 and Table 3), corroborating the lower sensitivity of the TbAT1-KO and B48 strains to **9** (Table 2 and Fig. 4). This indicates a high level of reliance of **9** on P2 transport and is in line with the identification of the encoding gene (*TbAT1*) in the whole-genome RNAi screening experiment. Interestingly, the addition of a bromine at the 7-position of **9** (analogue **10**) improved the affinity for P1 10-fold and recovered

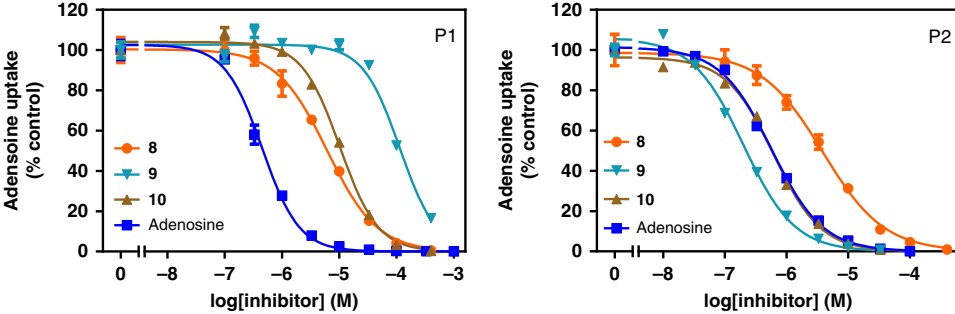

**Fig. 5** Transport of [³H]adenosine via P1 and P2 transporters in the presence nucleoside analogues. Uptake of [³H]adenosine was measured in the presence of increasing concentrations of nucleoside analogues **8**, **9** and **10**. Inhibition plots of adenosine and its analogues versus P1- or P2-mediated [³H] adenosine transport. The graphs represent one representative out of three independent experiments ($n = 3$), each performed in triplicate. Error bars are SEM, which fall within the symbol when not shown. Source data are provided as a Source Data file.

**Table 3 Kinetic parameters nucleoside analogue for P1 and P2 transporters.**

| Compound | P1 transporter | | | P2 transporter | | |
|---|---|---|---|---|---|---|
| | $K_m$ or $K_i$ | $\Delta G^0$ | $\delta(\Delta G^0)$ | $K_m$ or $K_i$ | $\Delta G^0$ | $\delta(\Delta G^0)$ |
| Adenosine | **0.12 ± 0.02** | −39.4 | | **0.53 ± 0.02** | −35.8 | |
| 3 Cordycepin | 210 ± 48[a] | −21.0 | 18.4 | 0.18 ± 0.02[b] | −38.5 | −2.7 |
| 8 7-Br-TUB | 4.85 ± 0.83 | −30.3 | 9.1 | 1.99 ± 0.31 | −32.5 | 3.3 |
| 9 3'-Deoxy-TUB | 99.9 ± 21.7 | −22.8 | 16.6 | 0.149 ± 0.03 | −38.9 | −3.1 |
| 10 7-Br-3'-deoxy-TUB | 9.47 ± 1.4 | −28.7 | 10.7 | 0.502 ± 0.14 | −35.9 | −0.1 |

$K_m$ (highlighted in bold) and $K_i$ values are in μM; $\Delta G^0$ in kJ mol⁻¹. $K_m$ and $K_i$ values are mean ± SEM, derived from three independent experiments ($n = 3$) with three replicates. $\delta(\Delta G^0)$ was calculated in comparison to adenosine. Source data are provided as a Source Data file.
[a]Value was taken from ref. [29]
[b]Value was taken from ref. [18]

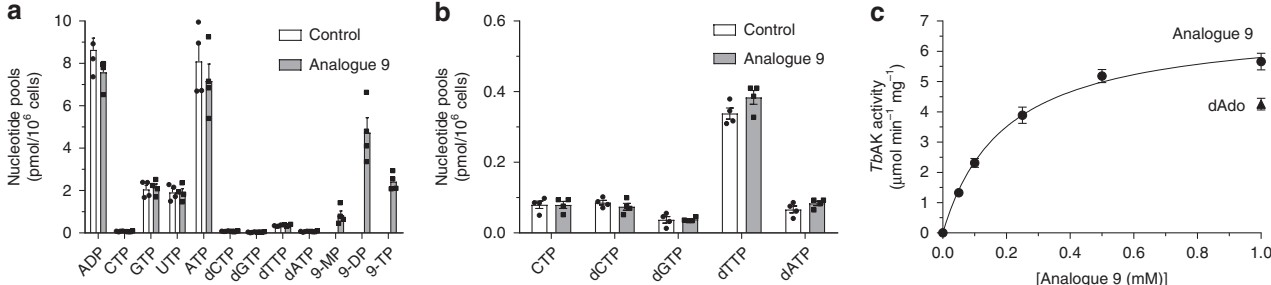

**Fig. 6** Investigation of the intracellular metabolism of **9**. **a** Intracellular (deoxy)nucleotide levels after exposure to analogue **9** (black bars), compared to non-treated control cells (white bars). **b** Enlargement of less abundant nucleotides from **a**. **c** Determination of kinetic parameters of **9** incubated with purified recombinant *Tb*ADKIN enzyme. The graphs represent the average of 3–4 independent experiments with the error bars indicating SE ($n = 4$ for **a**, **b** and $n = 3$ for **c**). Source data are provided as a Source Data file.

5.9 kJ mol⁻¹ in Gibbs free energy (Table 3), whereas the presence of the 3'-OH (**8** versus **10**) only doubled the affinity for P1 (Table 3). The very modest binding energy $\Delta G^0$ associated with the 3'-OH [$\delta(\Delta G^0) = 1.6$ kJ mol⁻¹], which has previously been shown to be one of the interaction points of adenosine with P1 having an apparent contribution of 18.4 kJ mol⁻¹ to the binding[29], appears to indicate that the 7-substituted analogues orient differently in the P1-binding pocket, making the 3'-OH less important.

The 3'-deoxy analogues displayed higher affinity for P2 than adenosine, with cordycepin (**3**) and **9** displaying almost identical $K_i$ values, roughly 3-fold lower than the $K_m$ value for adenosine, representing an energy advantage of ~3 kJ mol⁻¹, the same as the $\delta(\Delta G^0)$ for **10** compared to **8** (Table 3). This is similar to the energy gain upon removal of the 2'-OH[29] and consistent with P2

being essentially an adenine transporter condoning the ribose moiety[31,32].

**Intracellular fate of analogue 9.** RNAi experiments revealed the importance of *ADKIN* for the activity of nucleoside **9**. Therefore, the intracellular nucleotide pools of *T. brucei* after exposure to 25 μM of this nucleoside analogue were further analysed. Compound **9** did not cause any change in the balance or total quantity of intracellular purine and pyrimidine nucleotides in comparison to untreated trypanosomes (Fig. 6a, b). However, the analysis unambiguously showed that **9** is internalized and metabolized by *T. brucei* to mono-, di- and triphosphates (Fig. 6a, Supplementary Figs. 7 and 8). Incubation of analogue **9** with purified recombinant *Tb*ADKIN was performed to characterize the kinetic parameters of the first phosphorylation step (Fig. 6c), revealing a

$K_m$ of $195 \pm 24\,\mu\text{M}$ (mean and SE, $n = 3$) and $V_{max}\,\text{mg}^{-1}$ of $0.116 \pm 0.0048\,\text{U}$ ($\mu\text{mol s}^{-1}$) (mean and SE, $n = 3$), similar to what is reported for 2′-deoxyadenosine[33]. In agreement, we got comparable activities with 1 mM **9** and 2′-deoxyadenosine that was used as control (Fig. 6c).

**Analogue 9 does not impact *T. brucei* DNA and RNA synthesis.** Cell cycle analysis by flow cytometry on *T. brucei* parasites exposed for 24 h to various concentrations of **9** indicated no major impact on DNA synthesis with only a minor effect on relative trypanosome numbers in the S phase (Fig. 7), suggesting

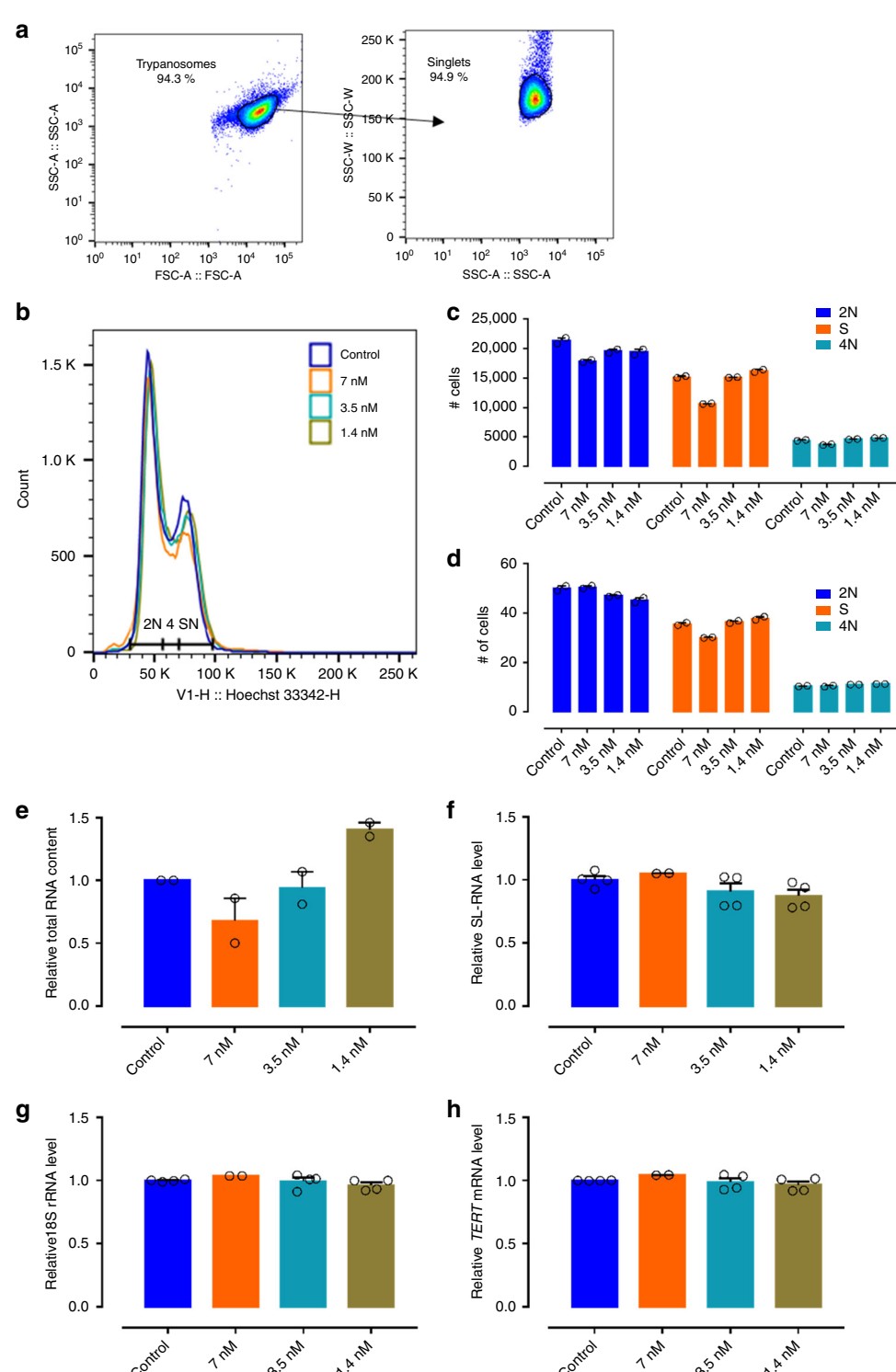

**Fig. 7** Evaluation of the effect of **9** on cell cycle regulation and RNA synthesis. **a** Gating strategy. **b** Results of the cell cycle analysis at different concentrations of compound **9**. Results of the cell cycle analysis (2*N*, S phase, 4*N*) expressed in total cell numbers (**c**, $n = 2$) and relative cell numbers (**d**, $n = 2$). **e** Total RNA yield from trypanosomes exposed to different concentrations of compound **9** ($n = 2$). RNA content quantified using RT-qPCR specifically targeting SL-RNA (**f**, $n = 2$ with two replicates), 18S rRNA (**g**, $n = 2$ with two replicates) and *TERT* mRNA (**h**, $n = 2$ with two replicates). RNA content and transcript levels are expressed as relative to the non-treated trypanosomes (control). Error bars are SEM. Source data are provided as a Source Data file.

that the formed triphosphate of analogue **9** (**9-TP**) does not inhibit the parasite's DNA polymerase. Subsequently, the impact of various concentrations of analogue **9** on RNA synthesis was analysed using standard quantification methods on the total RNA extracts and specific quantitative reverse transcription PCR (RT-qPCR) for the quantitative detection of *T. brucei* SL-RNA, 18S ribosomal RNA (rRNA) and *TERT* (telomerase reverse transcriptase) transcripts (Fig. 7e–h). The results indicate that neither nucleoside **9** nor any of its derived phosphorylated metabolites act as chain terminators for RNA synthesis given the absence of a major impact on the total RNA pool and the levels of specific transcripts produced by RNA polymerase I and II.

## Discussion

For decades nucleoside (and nucleobase) analogues have been successfully used in the clinic, particularly for the treatment of viral infections and cancer, which is evidenced by the more than 40 chemical entities from this compound class that are Food and Drug Administration approved at the present date[34,35]. Their utilization in the treatment of parasitic infections, however, is currently limited to the occasional use of allopurinol against canine leishmaniasis[36], although many other purine and pyrimidine analogues have been tested for their anti-trypanosomal and leishmanicidal effects, most prominently against *T. brucei*[14,17,18,21,25,37]. Using nucleoside analogues as chemotherapeutic agents against HAT potentially benefits from the inherently higher likelihood of crossing the BBB than for most other water-soluble compounds[22], thus enabling their use in the neurological stage of the disease[1].

By combining the molecular scaffolds of two well-studied antitrypanosomal purine nucleoside analogues, tubercidin and cordycepin, we have discovered 3′-deoxy-7-deazaadenosine derivatives with highly potent trypanocidal activity, validating the use of focussed purine nucleoside libraries as an effective strategy in phenotypic drug discovery. The frontrunner analogue **9**, which was first described in 1973 by Robins et al.[38], and later assessed by researchers at Merck for its in vitro activity against hepatitis C virus[39], has never been evaluated as a trypanocide.

A major issue for the therapeutic potential of adenosine-like nucleoside analogues is the high serum adenosine deaminase activity in vivo, resulting in the formation of potentially inactive inosine-like nucleosides, which is most prominently exemplified by cordycepin, requiring co-administration with 2′-deoxy-coformycin to ensure sufficient in vivo activity[17,18,23]. The absence of N7 in tubercidin-derived analogues potentially circumvents this issue[40], obviating co-administration. Analogue **9** was metabolically stable and showed remarkable efficacy in animal models of both stage 1 and stage 2 HAT at doses as low as 6.25 mg kg$^{-1}$ s.i.d. after oral gavage.

In their search for host purines, *T. brucei* parasites express several purine (nucleoside) transporters[41], coupled to an elaborate purine salvage pathway network enabling them to grow on virtually every purine source[14,42]. Following the uptake of adenosine, two metabolic routes are possible, which entails either a cleavage step (deribosylation by means of inosine-adenosine-guanosine nucleoside hydrolase[43], or methylthioadenosine phosphorylase[44]) or the direct phosphorylation by ADKIN to AMP[33,45]. We show here that, similarly to adenosine, analogue **9** is phosphorylated by *Tb*ADKIN, although kinetic analysis revealed it to be a low-affinity substrate of the recombinant enzyme. We therefore hypothesize that **9** accumulates in the trypanosomal cell, driven by the active uptake that coupling of the *T. brucei* purine transporters to the proton-motive force engenders[46,47], at which point it becomes a high capacity substrate (with a $V_{max}$ similar to that of cordycepin[33]).

The absence of major perturbations in the nucleotide pool of cells treated with **9** appears to indicate that it exerts its trypanocidal activity by some mechanism other than causing an imbalance in the intracellular nucleotide pool[21]. Moreover, the absence of DNA fragmentation or disruption of DNA synthesis shows that **9** is not incorporated into DNA, as the 3′-deoxy analogue would be an obligate chain terminator. Additionally, we have shown that **9** (or any of its phosphorylated metabolites) does not interfere with RNA synthesis. The presence of all three phosphorylated metabolites, as shown by high-performance liquid chromatography (HPLC) analysis, further complicates mode-of-action investigations, as it was shown that the triphosphate of tubercidin was responsible for the inhibition of *T. brucei* phosphoglycerate kinase[24], an enzyme from the glycolytic pathway.

In summary, we here described the discovery and effects of potent antitrypanosomal agents having a 3′-deoxy-7-deazaadenosine structure. Analogue **9** represents a potent and orally bioavailable chemotherapeutical candidate for treatment of human and animal African trypanosomiases, which shows no detrimental level of cross-resistance with currently used drugs.

## Methods

**Ethics statement for animal models.** The use of laboratory rodents was carried out in strict accordance to all mandatory guidelines (EU directives, including the Revised Directive 2010/63/EU on the Protection of Animals used for Scientific Purposes that came into force on 1 January 2013, and the Declaration of Helsinki in its latest version) and was approved by the Ethical Committee of the University of Antwerp, Belgium [UA-ECD 2017–04].

***Trypanosoma brucei* in vitro drug susceptibility assays.** Drug susceptibility tests with Lister 427WT, TbAT1-KO[48], B48[26] and ISMR1[27] were performed using an assay based on the viability indicator dye resazurin in 96-well plates, each well containing $2 \times 10^4$ cells. The plates were incubated for 48 h with a doubling dilution series of the test compounds in HMI-9/fetal bovine serum at 37 °C/5% CO$_2$ (23 dilutions starting at 100 μM, except for the pentamidine control (50 μM)), after which resazurin (20 μL of 125 μg mL$^{-1}$ resazurin sodium salt solution was added to each well (containing 200 μL of cells and test compound)) was added to each well and the plates incubated for another 24 h. Fluorescence was determined using a FLUOstar Optima (BMG Labtech, Durham, NC) and the results fitted to a sigmoid curve with variable slope using Prism 5.0 or Prism 7.04 (GraphPad, San Diego, CA).

Susceptibility assays with *T. brucei* Squib 427[49] or *T. b. rhodesiense* STIB-900[50] were performed under similar conditions as above, but using 10 concentrations of a 4-fold compound dilution series starting at 64 μM. *Trypanosoma brucei* Squib 427 was seeded at $1.5 \times 10^4$ parasites/well and *T. b. rhodesiense* at $4 \times 10^3$ parasites per well, followed by the addition of resazurin (final concentration of 10 μg mL$^{-1}$) after 24 h (*T. brucei*) or 6 h (*T. b. rhodesiense*).

Susceptibility assays with NY-SM[51], BS221[48] and TbAT1-KO[48] (derived from BS221) cells were performed using 10 concentrations of a 4-fold compound dilution series starting at 64 μM. Parasites were seeded at $4 \times 10^3$ parasites per well and exposed to drug compound for 72 h, after which resazurin (final concentration of 10 μg mL$^{-1}$) was added to the plates. Fluorescence intensities were determined after 6 h.

**Cytotoxicity on MRC-5 fibroblasts.** Cytotoxicity assays were performed on MRC-5$_{SV2}$ (Sigma-Aldrich/ECACC, catalogue number 84100401) human embryonic lung fibroblasts, which were cultured in minimum essential medium with Earle's salt medium, supplemented with L-glutamine, NaHCO$_3$ and 5% inactivated foetal calf serum. All cultures and assays were conducted at 37 °C with 5% CO$_2$. Ten microliters of the aqueous compound dilutions were added to 190 μL of MRC-5$_{SV2}$ ($3 \times 10^4$ cells mL$^{-1}$). Cell growth was compared to untreated control wells (100% cell growth) and medium control wells (0% cell growth). After a 3-day incubation, cell viability was assessed fluorimetrically after the addition of 50 μL resazurin per well (final concentration of 10 μg mL$^{-1}$). After 4 h at 37 °C, fluorescence was measured ($\lambda_{ex}$ 550 nm, $\lambda_{em}$ 590 nm). The results were expressed as percentage reduction in cell growth/viability compared to control wells and an EC$_{50}$ was determined.

**Mouse model of acute HAT (*T. b. brucei* Squib 427).** Female Swiss mice (body weight (BW) ~20–24 g; Janvier Labs, France) were allocated randomly to groups of three animals and infected intraperitoneally (i.p.) with 10$^4$ *T. b. brucei* Squib 427 derived from a heavily infected donor mouse. Drinking water and food were available ad libitum throughout the experiment. Analogue **9** was formulated in 10%

(v/v) polyethylene glycol 400 in water at 2 mg mL$^{-1}$ or in 5% (v/v) Tween-80 in water at 4 mg mL$^{-1}$ and was freshly prepared at every administration. Analogue **9** was administered orally b.i.d. for 5 days at 25, 12.5 and 6.25 mg kg$^{-1}$ or i.p. s.i.d. for 5 days at 10 mg kg$^{-1}$ (all groups $n = 3$). The reference drug suramin was formulated in phosphate-buffered saline (PBS) at 2.5 mg mL$^{-1}$ and administered s. i.d. i.p. for 5 days at 10 mg kg$^{-1}$ ($n = 3$). All treatments were initiated 30 min prior to the i.p. infection. Animals were observed for the occurrence of clinical or adverse effects during the course of the experiment and were weighed daily. Parasitaemia was determined by microscopic evaluation of tail vein blood samples at 4, 7, 10, 14 and 21 dpi (pre-set endpoint). As a test of cure, blood samples (250 μL) were collected from treated mice at 21 dpi and were sub-inoculated i.p. in naive Swiss mice ($n = 3$), followed by monitoring of parasitaemia as follow-up.

**Mouse model of stage 2 HAT.** Female Swiss mice (BW ~ 20–24 g; Janvier Labs France) were randomly allocated to groups of 4 animals and infected i.p. with 10$^4$ *T. b. brucei* AnTAR1.1 PPYRE9[52] derived from a heavily infected donor mouse. Analogue **9** was formulated in 5% (v/v) Tween-80 in water at 4 mg mL$^{-1}$ and was freshly prepared at every administration. Analogue **9** was administered by oral gavage for 5 days at 25 mg kg$^{-1}$ b.i.d. ($n = 2$), 25 mg kg$^{-1}$ s.i.d. ($n = 3$), 12.5 mg kg$^{-1}$ s.i.d. ($n = 2$) or 6.25 mg kg$^{-1}$ s.i.d. ($n = 3$). The reference drug melarsoprol was formulated as a stock solution of 3.6% in propylene glycol. The stock solution was converted into a gel by the addition of hydroxypropylcellulose to a final concentration of 1.5%. Melarsoprol was administered topically s.i.d. for 3 days at ≥120 mg kg$^{-1}$ ($n = 2$). Treatment was initiated at 21 dpi. Parasitaemia was determined microscopically using a Neubauer improved haemocytometer at 21, 25, 28, 32, 35, 43, 50, 59 and 63 dpi. Bioluminescent images were acquired at 7, 21, 28, 35, 43, 50, 59 and 63 dpi to evaluate parasite burdens in the various organs. Mice were injected i.p. with 15 mg kg$^{-1}$ of D-luciferin and anaesthetised using isoflurane. The luminescent signal was acquired from both the dorsal and ventral side of the animals. Bioluminescent images were acquired over an exposure time of 5 s from animals prior to treatment (until 3 wpi) and from animals in the vehicle control ($n = 2$) group that harbour high parasite burdens. Animals subjected to the different treatment schemes (from 4 wpi) were additionally imaged over a 5 min exposure time for enhanced sensitivity. Surviving animals at the 63 dpi time-point were further monitored up to 98 dpi. At 121 dpi, mice were sacrificed and brain, fat and spleen samples were collected for qPCR analysis to determine the presence of residual parasite burdens. Tissues were weighed and RNA was extracted using the RNeasy Plus Mini Kit (Qiagen) according to the manufacturer's recommendations. Parasite levels in these tissues were determined using quantitative real-time PCR targeting SL-RNA as described previously[28]. Quantitative real-time PCR targeting the eukaryotic translation elongation factor 2 (*Eef2*), a mouse reference gene, was performed in parallel, to confirm appropriate RNA extraction in all tested samples[53]. A list of primers used is provided in the Supplementary Methods section of the Supplementary Information. BLI images from the IVIS Spectrum In Vivo Imaging System (Perkin Elmer) were analysed using LivingImage v4.3.1. SoftWoRx suite 2.0 for microscopy (image deconvolution), then processing using the Fiji software.

**Whole-genome RNAi screening.** The RNAi library[54] was induced for 68 h by the addition of 1 μg mL$^{-1}$ of tetracycline. After induction, the cells were exposed to different concentrations of **9** (corresponding to the IC$_{98}$, IC$_{99}$). Cells were maintained until a recovery of the resistant population was observed. Genomic DNA of the resistant parasite population was isolated, and the RNAi inserts were identified by PCR amplification, followed by cloning in highly competent *Escherichia coli* and sequencing. To validate the involvement of the identified genes, the RNAi inserts were back-cloned into a p2T7Bern vector[54] and transformed into NY-SM cells (the background strain of the RNAi library). The transformed NY-SM cells were cloned by limiting dilution. Two clones of each RNAi insert were exposed to tetracycline to induce the RNAi phenotype and subsequently subjected to different concentrations of **9** to determine the susceptibility of induced versus non-induced clones.

Drug susceptibility assays with induced clones were performed as described above for NY-SM cells, with an initial induction period of 68 h with 1 μg mL$^{-1}$ tetracycline prior to the start of the drug susceptibility assay.

**In vitro adenosine transporter assay.** Transport via P1 was measured using B48 cells, which lack the P2 transport system[26], using a [$^3$H]adenosine concentration of 0.1 μM, whereas the transport via P2 was assessed in B48 cells transfected with *TbAT1*(P2) gene (B48 + TbAT1)[31] for a constant, high level of expression, in the presence of 100 μM of inosine to block the P1 transporter (0.033 μM [$^3$H]adenosine). The transport of [$^3$H]adenosine (40 Ci mmol$^{-1}$; American Radiolabelled Chemicals, St. Louis, MO) was measured using the following uptake protocol:[21,55] trypanosomes ($1 \times 10^7$ cells) were suspended in 100 μL assay buffer (33 mM HEPES, 98 mM NaCl, 4.6 mM KCl, 0.55 mM CaCl$_2$, 0.07 mM MgSO$_4$, 5.8 mM NaH$_2$PO$_4$, 0.3 mM MgCl$_2$, 23 mM NaHCO$_3$, 14 mM glucose, pH 7.3)[55] at room temperature and incubated with 100 nM [$^3$H]adenosine in the same buffer for 60 s, followed by rapid termination by the addition of ice-cold 2 mM adenosine, followed by immediate centrifugation in a microfuge through an oil

layer (250 μL of 7:1 mixture of di-*n*-butylphthalate and light mineral oil (Sigma)) for 1 min at 13,000 r.p.m. Non-specific association of radiolabel with the cell pellet was determined using an identical incubation in the presence of saturating (1 mM) unlabelled adenosine. The incubation times used were well within the linear phase of uptake[47]. Inhibition constants ($K_i$) were calculated from 50% inhibition values (IC$_{50}$) obtained by non-linear regression (sigmoid curve with variable slope; GraphPad Prism 5.0) and the Cheng–Prusoff equation $K_i = IC_{50}/(1+(L+K_m))$, in which $L$ is the permeant concentration[55]. The Gibbs free energy was calculated from $\Delta G^0 = -RT\ln(K_i)$ as described[29], in which $R$ is the gas constant and $T$ the absolute temperature. Prism 7 (GraphPad) was used for the analysis of transporter data, fitting to a sigmoid dose–response curve with variable slope.

**Nucleotide pool analysis.** Logarithmically growing *T. brucei* Lister 427 cells (50 mL, $1.5–2 \times 10^6$ cells mL$^{-1}$) were incubated for 1 h with compound **9** (25 μM final concentration) added to the growth medium at 37 °C. Afterwards, the cell-containing flasks were placed on ice for 5 min before harvesting by centrifugation for 5 min ($4000 \times g$) at 4 °C. The supernatant was discarded and the cells were gently resuspended and transferred to an Eppendorf tube. The sample was centrifuged for 1 min at maximum speed ($21,000 \times g$), and the pellet was resuspended in 50% (v/v) acetonitrile by pipetting it up and down in the solution to disintegrate the cells. After re-centrifugation, the supernatant was centrifuged through a pre-washed Nanosep 3k Omega filter (PALL Corporation, Port Washington, NY, USA). The filtrate was evaporated in a speedvac, resuspended in water and stored at −20 °C and mixed 1:1 with mobile phase before HPLC analysis. Exact chromatographic conditions for each specific analysis are described in the Supplementary Information (Supplementary Figs. 7–11). Data for nucleotide pool analysis are based on four independent biological repeats.

***ADKIN* assay.** The *T. brucei* ADKIN enzyme was diluted in a buffer containing 0.05% (v/v) Tween-20, 50 mM KCl, 0.1 mM dithiothreitol, and 50 mM Tris-HCl, pH 7.5 (dilution buffer), prior to use in the assay. The final assay mixture contained 12.5 ng *T. brucei* ADKIN, a range between 50 and 1000 μM of nucleoside **9** (or 1000 μM deoxyadenosine in the control reaction), 100 mM KCl, 10 mM MgCl$_2$, 0.5 mM ATP, 5 mM potassium phosphate and 50 mM Tris-HCl, pH 7.5. The assay was incubated at 37 °C for 30 min, and subsequently heated for 2 min at 100 °C to inactivate the enzyme, diluted with 2 volumes of water, centrifuged through a Nanosep 3k Omega filter and mixed 1:1:2 with water and mobile phase. The HPLC analysis was performed on a $2.1 \times 150$ mm ACE Excel C18 column using a mobile phase composition 4A:76B:20C (see nucleotide pool analysis, Supplementary Information for details). Using these conditions, consecutive samples could be loaded (Supplementary Fig. 12). After 4–5 samples, the column was washed with high percent A (80A:20C) to remove bound ADP and ATP before re-equilibrating the column to original conditions.

**Cell cycle analysis.** NY-SM cells were exposed to different concentrations of **9** (corresponding to the IC$_{50}$, IC$_{50}$/2 and IC$_{50}$/5) for 24 h. After 24 h, the cells were harvested and washed with PBS before staining with Hoechst 33342 at 5 μg mL$^{-1}$ for 25 min at 37 °C. Cells were analysed on a MACSQuant flow cytometer (Miltenyi Biotec) using the FlowJo X software package.

**RNA quantification.** NY-SM cells were exposed to different concentrations of compound **9** (corresponding to the IC$_{50}$, IC$_{50}$/2, IC$_{50}$/5) for 24 h. After 24 h, the cells were harvested and washed with PBS. Extracts were made from a normalized number of cells, given that particularly incubation with compound **9** at the IC$_{50}$ concentration had a significant impact on the number of recovered trypanosomes. RNA was extracted using the QIAamp RNA Blood Mini Kit (Qiagen). Total RNA yields were determined using both Nanodrop 2000 spectrometry and Qubit fluorimetric RNA content analysis. Additionally, specific transcripts in the RNA pool were quantified using RT-qPCR targeting the SL-RNA, 18S rRNA and transcripts encoding the telomerase reverse transcriptase (*TERT*)[28,56]. A list of primers used is provided in the Supplementary Methods section of the Supplementary Information.

**Reporting summary.** Further information on research design is available in the Nature Research Reporting Summary linked to this article.

## Data availability
The authors declare that the data underlying the findings of this study are available within the paper and its Supplementary Information files. Genome sequence and annotation information was obtained from TritrypDB (http://www.tritrypDB.org). The source data underlying Figs. 3–7, Tables 1–3 and Supplementary Figs. 1, 2, 4, 5, 7 and Supplementary Table 1 and 2, are provided as Source Data File.

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

## Acknowledgements

F.H. is indebted to the FWO-Flanders for a Ph.D. scholarship. G.D.C. thanks Science Without Borders for his scholarship (206385/2014-5, CNPq, Brazil). G.C. is supported by a research fund of the University of Antwerp (TT-ZAPBOF 33049). The present work has been funded by the FWO (G.C., L.M., S.V.C.; project number G013118N). S.V.C. thanks Prof. emeritus Fred Opperdoes for the useful suggestion to collaborate with H.P.d.K. to study the involvement of adenosine transporters in the uptake of the nucleoside analogues. The authors express their gratitude to Dr. Jennifer Ann Black for assistance with the microscopy experiments. We thank Rik Hendrickx, Pim-Bart Feijens, An Matheeussen, Natascha Van Pelt, Mandy Vermont, Margot Desmet and Izet Karalic for excellent technical assistance.

## Author contributions

Design of research: F.H., A.H., H.P.d.K., G.C., S.V.C. Performed experiments: F.H., D.M., G.D.C., G.S., A.H. Analysed data: F.H., D.M., G.D.C., G.S., I.R., L.M., A.H., H.P.d.K., G.C., S.V.C. Contributed RNAi library: I.R., G.S. Wrote the manuscript: F.H., D.M., G.D.C., G.S., I.R., L.M., A.H., H.P.d.K., G.C., S.V.C. D.M. and G.D.C. contributed equally to the mode of action studies in this work.

## Competing interests

The authors declare no competing interests.
