## [Peer Review File · Nature Communications]

Reviewers' Comments:

Reviewer #1:

Remarks to the Author:

New agents targeting the purine salvage pathway of protozoan parasites have long been proposed but rarely achieved. In this breakthrough work, several purine analogues are shown to be effective in cell culture and in mice. The work is important and carefully executed. Publication is recommended after a few improvements to make it more accessible to readers:

1. Abstract – name the compound instead of “Analogue 9”
2. Abstract – name the “specific nucleoside transporters”
3. L 119 – jumps right into TbAT1/P2 and B48 without informing the reader what they are.
4. L 144-153 – 50% metabolism in 30 min is not very stable. The rest is confusing.
5. L 258 – Vmax best expressed as per sec.
6. L 293-299 – Your best evidence for drug efficacy is Fig 6A showing accumulation in cells.

Reviewer #2:

Remarks to the Author:

This is a very nice study. Relatively simple nucleoside analogs (known before to treat other infectious diseases) show good potency to essentially cure rodents of stage I and stage II HAT. Dosing down to ~6 mg/kg achieves cures, which is where you want to be to move this compound toward a clinical candidate. I think the paper is important as new drugs to treat HAT are needed, and the authors show a promising set of compounds that could advance to clinical trials.

My only issue is for the authors to show that the compounds are cytotoxic rather than just cytostatic. They talk about this in terms of the rate of trypanocidal effects but it is not a rate issue. The authors should incubate parasites in vitro with compound above the EC50 and then wash away the compound, do the parasites come back after further culture, if so it is static, if they don't, it is cidal.

Reviewer #3:

Remarks to the Author:

This communication reports that combination of the scaffolds of tubercidin and cordycepin results in a compound (9) with activity against a rodent model of late stage sleeping sickness. Both of these adenosine analogs were discovered over 50 years ago and their trypanocidal activity has been known since the 1960s, (please cite references in review by J. Williamson, Review of Chemo-therapeutic and chemoprophylactic agents in the African Trypanosomiases). Toxicity and metabolic instability have been major limitations to their therapeutic use and this report provides evidence that 3-deoxy-7-deazadenosine (compound 9) may have overcome these limitations. The authors also investigate the mode of uptake and activation of this pro-drug but fail to elucidate its mode of action. The manuscript is clearly and concisely written but would benefit from revision to address the following.

Major points

1. Efficacy in rodent models. The fixed end-point for assessment of cure in most studies of African trypanosomiasis are a parasitaemic survival for >30 days or >200 days in the acute and chronic mouse model, respectively. The authors have used a parasitaemic survival >21 days and >63 days for their stage I and stage II *T. brucei brucei* infection studies and absence of parasitaemia at 14 days post-infection in their *T. congolense* model. What is the evidence to support these endpoints as indicators of cure?

2. Treatment regimen. It is not clear how long after infection that treatment was commenced in the stage I model. Lines 366-367 states "Treatment was initiated half an hour prior to IP infection". It is not clear whether this refers solely to suramin (IV route) or also to compound 9. In any case, this is not a suitable protocol to assess cure of infection rather than determining prophylactic activity.

In the stage II model (line 373), the number of mice used in the study is not given (Figure 3 would suggest $n = 2$). If so, what is the justification for $n = 2$? Is this sufficiently powered? Also, what is the limit of detection for parasites in the brain? How can one be sure that this equates to cure given relapse reported in other published CNS models occur beyond day 63?

3. What is the therapeutic index for compound 9? Reference 36 indicates that the closely related compound 6 (7-deaza-adenosine) is lethal at 50 mg/kg, but tolerated with no visible signs of toxicity at 20 mg/kg over 14 days. If the maximum tolerated dose is similar, then 9, which is curative at 6.25 mg/kg for 5 days, may only have a low therapeutic index of about 8. It is a deficiency of this study that an ED50 and ED90 value for stage II cure (survival >200 days) has not been determined.

Minor points

1. Potency data would be easier to follow in the figures, tables and text if given in nM units.
2. Line 66. This is an overestimate. Post-treatment reactive encephalopathy occurs in 5-10 % of patients with a fatality rate of 50%, i.e. 2.5 – 5%, not 6-9% deaths.
3. Lines 69-70. Suramin is also given intravenously and pentamidine intramuscularly. The authors could make it clearer that, with the exception of fexinidazole, all current treatments suffer from the requirement for parenteral administration.
4. Line 72. I suggest adding to the end of this sentence ... against both stage I and stage II HAT.
5. Figure 2. This should be divided into a separate figure and a table. For clarity, EC50-values would be better given in nM units to 2-significant figures. Please determine an EC50 value for compound 9 with STIB-900 T. rhodesiense.
6. Lines 181-187. The paper could be shortened and more focussed if the preliminary data with T. congolense was deleted. As mentioned above the end-point for curative activity in vivo is questionable and without similar biological studies on the other key AAT infectious agent T. vivax, it cannot be concluded that 9 is a promising lead for animal trypanosomiasis. Furthermore, the target product profile is different (e.g. intramuscular or subcutaneous route for AAT rather than oral route for HAT).
7. Line 238-239. If the P2-transporter is essentially an adenine transporter condoning the addition of a ribose moiety, then what is the reason for the poor cellular activity of 11? Is it a poorer substrate than 9 for phosphorylation by adenosine kinase?
8. Line 314. Can you clarify why 9 would not act as an obligate chain terminator for RNA synthesis?
9. In vitro drug susceptibility assays. Please specify the final concentration of resazurin used in the assays.
10. Line 438. Units should be micromolar?
11. General stylistic points. Use SI units for time consistently in the text. Do not start sentences with a

number (lines 363, 364, 375, 376, 414 and elsewhere). References 7, 9, 11, 19, 25, 26, 28, 30, 39-45 lack italics for species names or incorrect capitalization.

12. Table 1. Data for *T. congolense* is incomplete and as mentioned above, I suggest removing any reference to animal African trypanosomiasis.

13. Supplementary, page 3. Is it 6 or 8 that has a trypanostatic effect?

14. Supplementary figure S2. The images are rather small to confirm the morphological changes. Consider an alternative form of presentation.

15. Supplementary Figure S6. This should be presented as a Table. It is not clear why two *T. b. gambiense* genes appear in this list since the RNAi experiment involved *T. b. brucei*.

16. Statistics. Number of animals used per group is not clear. Figure 4, P-values are not exact and the statistical method is not specified.

Reviewer 1:

- 1. Abstract – name the compound instead of “Analogue 9”
 - ‘analogue 9’ has been replaced by 3’-deoxytubercidin.

- 2. Abstract – name the “specific nucleoside transporters”
 - ‘specific nucleoside transporters’ has been replaced by ‘P1 and P2 nucleoside transporters’.

- 3. L 119 – jumps right into TbAT1/P2 and B48 without informing the reader what they are. This was changed and now reads: “...lacking the *TbAT1* gene encoding the P2 aminopurine transporter, i.e. TbAT1-KO and clone B48 (derived from TbAT1-KO, with additional loss of the HAPT1/AQP2 transporter).”

An additional reference has been added to clarify the B48 trypanosomal cell line.
Reference (ref.#47 in the originally submitted manuscript): Bridges, D. J. et al. *Mol. Pharmacol.* **2007**, 71 (4), 1098-1108.

- 4. L 144-153 – 50% metabolism in 30 min is not very stable. The rest is confusing
 - We agree that our phrasing in L144 could have been better. We do think that derivative **9** is metabolically stable, as we consider 50% parent compound remaining at 30 min of incubation the cut-off value for making this statement, and therefore this was mentioned in the text.

In the revised manuscript, we have removed this latter part in line 144-146; which now reads: “Incubations with mouse, rat and human S9 microsomal fractions revealed that **9** is metabolically stable, defined as $\geq 50\%$ of parent compound remaining after 30 min (Supplementary Table S1).”

It is our opinion that the information about the susceptibility against (serum) adenosine deaminase is of importance, as this is a difference between analogue **9** and cordycepin (analogue **3**), which hampers the use of the latter as a trypanocide (see e.g. Vodnala et al *J. Med. Chem.* **56**:9861-9873 (2013)).

We have reformulated this paragraph to present this information more clearly. We think it is necessary to present this information on adenosine deaminase in this paragraph, as deamination represents a metabolic conversion typical for certain adenosine(-like) nucleoside analogues, but is not addressed in the microsomal stability assays, as adenosine deaminase is a serum protein.

Additionally, in this paragraph (see comment from rev.#3), we have deleted the part that refers to *T. congolense*.

- 5. L 258 – Vmax best expressed as per sec.
 - The original value of V_{\max}/mg of $6.93 \pm 0.29 \text{ U } (\mu\text{mol} \cdot \text{min}^{-1})$ has been changed to $0.116 \pm 0.0048 \text{ U } (\mu\text{mol} \cdot \text{s}^{-1})$.

- 6. L 293-299 – Your best evidence for drug efficacy is Fig 6A showing accumulation in cells.
 - We thank rev.#1 for this remark.

Reviewer 2:

- My only issue is for the authors to show that the compounds are cytocidal rather than just cytostatic. They talk about this in terms of the rate of trypanocidal effects but it is not a rate issue. The authors should incubate parasites in vitro with compound above the EC50 and then wash away the compound, do the parasites come back after further culture, if so it is static, if they dont, it is cidal.
 - We would like to argue that Supplementary Figure S1 is indicative of a cidal effect. The observed effects on the growth curve show that the total number of parasites present at different time levels is decreasing upon treatment with the 3'-deoxy analogues **9** and **10**, whereas for analogue **8** (7-bromotubercidin), this remains similar to the original seeding density (enlargement in panel B). This observation can only be explained by a direct cidal effect of analogues **9** and **10** (note that the non-treated cells have significant growth over the same period; Panel A).

Nonetheless, we have performed the experiment as suggested by reviewer #2 (results and experimental details have been added in the Supplementary information (new Figure S2)). Incubation of cell cultures at 0.165 μM or 0.50 μM for analogue **9**, showed that analogue **9** is indeed trypanocidal, since at a concentration of 0.500 μM no parasites were recovered after 48h of exposure and subsequent removal of compound from the medium; no trypanosomes could be detected up to the end of the experiment at 120 h. Removal of the drug after 8 h had apparently not resulted in a complete killing of all parasites, as trypanosomes could be detected again at the 96 h and 120 h points. The clear conclusion is that the compound is rapidly trypanocidal, leading to a complete sterilisation of the culture in >8 h and <48 h of exposure to a submicromolar concentration of analogue **9**.

Reviewer 3:

- This communication reports that combination of the scaffolds of tubercidin and cordycepin results in a compound (**9**) with activity against a rodent model of late stage sleeping sickness. Both of these adenosine analogs were discovered over 50 years ago and their trypanocidal activity has been known since the 1960s, (please cite references in review by J. Williamson, Review of Chemo-therapeutic and chemoprophylactic agents in the African Trypanosomiases).
 - We thank rev.#3 for this comment. Two additional references, which are the first reports on the activity of cordycepin and tubercidin, respectively, against African trypanosomes, have been added:
 - a) Williamson, J., *Trans. R. Soc. Trop. Med. Hyg.* **1966**, *60* (1), 8-8.
 - b) Williamson, J., *Trans. R. Soc. Trop. Med. Hyg.* **1969**, *63* (4), 422-423.

Major points:

- 1. Efficacy in rodent models. The fixed end-point for assessment of cure in most studies of African trypanosomiasis are aparasitaemic survival for >30 days or >200 days in the acute and chronic mouse model, respectively. The authors have used aparasitaemic survival >21 days and >63 days for their stage I and stage II *T. brucei brucei* infection studies and absence of parasitaemia at 14 days post-infection in their *T. congolense* model. What is the evidence

to support these endpoints as indicators of cure?

In order to respond to this remark made by rev.#3, we have separated the remark into three distinct subparts, all which have been answered separately.

➤ Acute stage model

In order to support cure in the stage-I model, we have collected blood from surviving animals at 21 dpi, and inoculated them in treatment-naïve mice. Follow-up of these animals also resulted in the absence of parasitaemia, thus supporting sterile cure. This experiment was mentioned in the original manuscript (Methods section) and has now been specified with number of mice and administration route.

➤ Stage II model

The stage II model makes use of bioluminescent parasites (*i.e.* constitutively expressing a luciferase enzyme), which allows to track low levels of parasites in animals by means of BLI imaging after injection with the substrate D-luciferin.

We have made use of virtually the same protocol as described by J. C. Mottram and co-workers; reference: Myburgh, E. et al., *PLoS Neglected Trop. Dis.* **2013**, 7 (8), e2384.

Similarly to this protocol, we have initiated treatment of mice at 21 dpi. Typically, in this animal model, luminescence in the CNS region (due to relapse) becomes visible in the period between 41 dpi to 51 dpi (derived from experiments with diminazene and DB75), which validates the use of 63 dpi as a suitable end-point. This animal model, employing luciferase-positive parasites allows for the faster assessment of cure and the use of lower numbers of laboratory animals, which are the main reasons for implementing this technology, particularly the time-reduction.

In fact, we have performed additional follow-up on surviving mice by BLI (Supplementary Figure S5 has now been altered) for animals receiving 25 mg/kg bid, 25 mg/kg sid, 12.5 mg/kg sid and 6.25 mg/kg sid, in parallel with the melarsoprol control. The follow-up was extended to 14 wpi (*i.e.* 98 dpi) by means of BLI monitoring (an additional row of figures displaying the BLI images at 14 wpi has been added to Supplementary Figure S5). This showed no signs of relapse, as derived from the absence of BLI signal in these animals.

Furthermore, surviving animals (which received 25 mg/kg bid, 25 mg/kg sid, 12.5 mg/kg sid and 6.25 mg/kg sid, as well as the melarsoprol control) we sacrificed at 121 dpi, and brain, spleen and fat tissue samples were collected. These were then subsequently analysed by rt-qPCR. This employed as a marker for trypanosomal RNA, the spliced-leader RNA (SL-RNA). Concurrently, in order to make sure RNA extractions were adequately performed and that no significant sample degradation had occurred, the mouse Eef2 (eukaryotic translation elongation factor) was assayed. These experimental data have been added to the Supplementary information (Supplementary Table S3). The methods sections has been altered to describe these additional experiments and the following sentence has been added to the relevant main text section: 'Additional follow-up of surviving mice with BLI up to 14 wpi did not show any signs of relapse. All treated (melarsoprol and analogue 9) animals

showed complete absence of parasite burdens as assayed in several tissue samples (brain, spleen and fat) by a highly sensitive SL-RNA qPCR detection method (Supplementary Table S3)'.

In view of these data, we conclude that the animals were completely devoid of any residual parasites, therefore supporting our claim of cure.

Additionally, we would like state that, to the best of our knowledge, currently no research group that makes use of BLI technology to monitor parasitic infections for African trypanosomes has been able to accurately determine a specific limit of detection for parasites in the brain area. However, there is a report of the limit for the intraperitoneal space being as low as 100 parasites. (McLatchie, A.P. et al. *PLoS Neglected Trop. Dis.* **2013**, *7* (11), e2571.

➤ *T. congolense* model

We agree with the comment of rev.#3 (see below) that the data on *T. congolense* are indeed premature in order to firmly claim that analogue **9** has potential for the treatment of AAT. Therefore, as suggested by rev.#3 we have removed all information concerning AAT and *T. congolense*, particularly as given the time frame that was provided for to produce this revision, we were unable to perform the key animal experiments with *T. vivax* in order to fully support our statement of analogue **9** being an analogue of interest for AAT.

- 2. Treatment regimen. It is not clear how long after infection that treatment was commenced in the stage I model. Lines 366-367 states "Treatment was initiated half an hour prior to IP infection". It is not clear whether this refers solely to suramin (IV route) or also to compound **9**. In any case, this is not a suitable protocol to assess cure of infection rather than determining prophylactic activity.

➤ This has been clarified, with the sentence now reading 'All treatments were initiated 30 min prior to the i.p. infection'.

➤ The first dose of treatment is indeed administered 30 min before the i.p. infection of the mice in the acute model. Since a monomorphic strain of *T. brucei* is used, which quickly results in high levels of blood parasitaemia (visible for the non-treated animals that develop blood parasitaemia $>10^8$ parasites/mL by 4dpi), the choice was made to administer a dose shortly before initiating the infection.

We would agree with rev.#3 that this model of infection is not very stringent. However, the use of the CNS-stage animal model, which is the key experiment in this study, is a stringent model in which compound **9** has shown to deliver a cure of an established and indeed advanced infection.

- In the stage II model (line 373), the number of mice used in the study is not given (Figure 3 would suggest $n = 2$). If so, what is the justification for $n = 2$? Is this sufficiently powered? Also, what is the limit of detection for parasites in the brain? How can one be sure that this equates to cure given relapse reported in other published CNS models occur beyond day 63?

➤ Animal experimental studies were conducted with a lower number of animals ($n = 2-4$ /group) than calculated by our standard power analysis, which had suggested the use of 6 mice/group (2-sample t-test, power= 80%, $\alpha = 0.05$). Ethical considerations, combined with experience with these models and a relative low variability have been the basis of

justifying these reduced numbers. Observations were confirmed in an independent repeat experiment (Supplementary Figure S5).

The number of animals used in each treatment group has now been added.

Considering the concern raised by rev.#3 on the limit of detection in the brain, we would like to refer to our explanation given above and put forward that, to the best of our knowledge, it has never been described in the literature what the precise number of parasites is that still results in a measurable signal visible in the CNS region. However, as stated above, we have now included data in the manuscript (additional follow-up & qPCR data) providing additional support for the stated claim of cure.

- 3. What is the therapeutic index for compound 9? Reference 36 indicates that the closely related compound 6 (7-deaza-adenosine) is lethal at 50 mg/kg, but tolerated with no visible signs of toxicity at 20 mg/kg over 14 days. If the maximum tolerated dose is similar, then 9, which is curative at 6.25 mg/kg for 5 days, may only have a low therapeutic index of about 8. It is a deficiency of this study that an ED50 and ED90 value for stage II cure (survival >200 days) has not been determined.

➤ First, the authors would like to clarify that the study by Olsen, D. B. et al. (ref. 36 in the original manuscript; *Antimicrob. Agents Chemother.* **2004**, *48* (10), 3944-3953) examined acute toxicity of tubercidin (or 7-deazaadenosine, **6**) in mice after single oral dosing at 20 or 50 mg/kg.

In comparison, we have administered analogue **9** at a dose of 25 mg/kg b.i.d. for five consecutive days (highest dose assayed), thus potentially resulting in compound accumulation, without observing any compound-related side-effects.

We would also argue that a direct comparison with tubercidin to derive a therapeutic index is not justified, and this is in fact one of the major points of this work. We have shown that small modifications to the structure of nucleoside analogues not only potentially have a significant impact on *in vitro* potency, but also on *in vitro* cytotoxicity and on *in vivo* tolerability as well. This is equally so in the study of Olsen, D. B. et al. (ref. 36), where the addition of a single methyl group at the C2' position resulted in significantly better anti-HCV activity and good tolerability of the analogue, which was later nominated a clinical candidate.

We do agree with rev.#3 that a study designed to find the lowest (oral) dose(s) that is still able to elicit a pharmacological effect is of interest in follow-up studies. However, it is the author's belief that by providing evidence of compound efficacy at a range of doses and dosing regimens spanning 25mg/kg b.i.d. to 6.25 mg/kg s.i.d., this should suffice in providing a strong proof-of-concept. We would further like to remark that the doses administered to mice in our study, compared to efficacious doses found by other groups for other new antitrypanosomal agents, can be considered as being on the lower end. (e.g. Khare, S. et al. *Nature*, **2016**, *537* (7619), 229-233).

Finally, the authors would like to stress that the primary aim of this publication was to provide a strong proof-of-concept for analogue **9** as a potential new anti-sleeping sickness agent. We admit that a thorough toxicological evaluation of this analogue is warranted to

enable an IND-filing, but we do question the need to perform such experiments to support the conclusions made in this work.

Currently, we have not performed any experiments to determine the maximal tolerated dose, as this opposes current Ethical Guidelines regarding the use of Laboratory animals in place at the University of Antwerp (BE) where all animal experiments were conducted.

Minor points:

- 1. Potency data would be easier to follow in the figures, tables and text if given in nM units.
 - We have altered several values in the text to μM so that there they are all reported in the same fashion. We have chosen to keep μM values consistently as most researchers in the parasitology field are used to this concentration level.
- 2. Line 66. This is an overestimate. Post-treatment reactive encephalopathy occurs in 5-10 % of patients with a fatality rate of 50%, i.e. 2.5 – 5%, not 6-9% deaths.
 - We have altered these percentages in the text, according to the reviewer's suggestion, and added additional references.

References added:

Pépin, J. et al., *Trans. R. Soc. Trop. Med. Hyg.* **1994**, *88* (4), 447-452.

Kennedy, P. G., *Ann. Neurol.* **2008**, *64* (2), 116-126.

- 3. Lines 69-70. Suramin is also given intravenously and pentamidine intramuscularly. The authors could make it clearer that, with the exception of fexinidazole, all current treatments suffer from the requirement for parenteral administration.
 - We thank rev.#3 for this suggestion, and have incorporated it. The sentence now reads:
“... necessity for parenteral administration (intravenous for suramin, melarsoprol and eflornithine and intramuscular for pentamidine),...” Oral fexinidazole is mentioned in the text immediately below this.
- 4. Line 72. I suggest adding to the end of this sentence ... against both stage I and stage II HAT.
 - We thank rev.#3 for this suggestion and have incorporated it.
- 5. Figure 2. This should be divided into a separate figure and a table. For clarity, EC50-values would be better given in nM units to 2-significant figures. Please determine an EC50 value for compound 9 with STIB-900 T. rhodesiense.
 - An EC₅₀ value for compound **9** was determined and presented in the original Figure. We assume that the question is in fact related to compound **6** (tubercidin) or compound **10** (3'-deoxy-7-bromoTubercidin), which we have now determined for both analogues and added to the table.
Compound **6**: EC₅₀ = 0.036 ± 0.001 μM
Compound **10**: EC₅₀ = 0.00040 ± 0.00009 μM

As suggested by rev.#3 we have split the original figure 2 into two parts; figure 2 and table 1. Additionally, we have now presented the in vitro data with two significant figures. However, we prefer to keep the μM unit for all drug sensitivity data, as this represents the most common format to report EC_{50} values in the parasitology field.

- 6. Lines 181-187. The paper could be shortened and more focussed if the preliminary data with *T. congolense* was deleted. As mentioned above the end-point for curative activity in vivo is questionable and without similar biological studies on the other key AAT infectious agent *T. vivax*, it cannot be concluded that **9** is a promising lead for animal trypanosomiasis. Furthermore, the target product profile is different (e.g. intramuscular or subcutaneous route for AAT rather than oral route for HAT).
 - As suggested, we have removed all reference to the *T. congolense* data. Our goal was to provide an additional proof-of-concept for activity against this veterinary parasite. However, we agree with rev.#3 that, without the data on *T. vivax*, we cannot yet confidently claim that analogue **9** is a promising lead for AAT. However, we are looking into the possibility of assessing potency in a *T. vivax* animal model, among other things, but this would now be the subject of a separate report.
- 7. Line 238-239. If the P2-transporter is essentially an adenine transporter condoning the addition of a ribose moiety, then what is the reason for the poor cellular activity of **11**? Is it a poorer substrate than **9** for phosphorylation by adenosine kinase?
 - We have not specifically measured the kinetic parameters for analogue **11** (2'-deoxytubercidin), but it is also our hypothesis that it being a poor substrate for adenosine kinase is the cause of the inactivity observed for this analogue; based on our knowledge of the P2 substrate selectivity, we anticipate that **11** should be a good substrate for this transporter. Nonetheless, given the difficulty in pinpointing the exact target of one (or all) of the phosphorylated metabolites that can potentially arise after reaching the intracellular environment of the parasite, there still exists the possibility that this analogue does perhaps not differ too much (in kinetic parameters for phosphorylation) but rather is unable to engage the same target (efficiently).
- 8. Line 314. Can you clarify why **9** would not act as an obligate chain terminator for RNA synthesis?
 - We have performed several experiments to investigate the impact of compound **9** on mRNA levels, the results of which have now been added to the manuscript (Figure 7). If compound **9** (or any / all of the phosphorylated metabolites) would impact RNA synthesis, one would expect to see a decline of total RNA pools. This has been assessed by Nanodrop 2000 spectrometry and Qubit fluorimetric RNA content analysis, none of which showed significant differences in total RNA pools, compared to non-treated control cells.

Furthermore, several RT-qPCR assays have been performed, specifically targeting *T. brucei* Spliced Leader (SL)-RNA, 18S rRNA and TERT transcripts. None of these experiments produced a significant difference between compound-treated vs. non-treated trypanosomes. This indicates that neither RNA polymerase I nor RNA polymerase II is affected by analogue **9** (or by the produced phosphorylated metabolites).

These data thus indicate that **9** is unable to act as an obligate chain terminator for RNA synthesis.

All data/methods to reach this conclusion have now been added to the manuscript.

- 9. In vitro drug susceptibility assays. Please specify the final concentration of resazurin used in the assays.
 - For experiments with strains: Lister 427WT, TbAT1-KO, B48 and ISMR1 cell lines. 20 μ L of 125 μ g/mL resazurin sodium salt was added to each well (already containing 200 μ L of cells and test compound), thus resulting in a concentration of 11.4 μ g/mL resazurin sodium salt (or 10.4 μ g/mL resazurin).
For experiments with strains: *T. brucei* Squib 427, NY-SM, BS221 and TbAT1-KO (derived from BS221) and *T. b. rhodesiense* STIB-900 : final concentration of 10 μ g/mL resazurin.
Cytotoxicity experiments with MRC-5 fibroblasts: final concentration of 10 μ g/mL resazurin.
This information has now been added to the methods sections.
- 10. Line 438. Units should be micromolar?
 - We thank rev.#3 for noticing this error. Indeed this unit should read μ M, and has been corrected. The units presented in Figure 6 Panel C are mM, which corresponds to the text in the methods section (first data point from compound added experiments is 50 μ M).
- 11. General stylistic points. Use SI units for time consistently in the text. Do not start sentences with a number (lines 363, 364, 375, 376, 414 and elsewhere). References 7, 9, 11, 19, 25, 26, 28, 30, 39-45 lack italics for species names or incorrect capitalization.
 - We have adjusted SI units and edited references (according to Nature publishing Guidelines).
- 12. Table 1. Data for *T. congolense* is incomplete and as mentioned above, I suggest removing any reference to animal African trypanosomiasis.
 - As indicated above, we have removed all data concerning *T. congolense*.
- 13. Supplementary, page 3. Is it 6 or 8 that has a trypanostatic effect?
 - We thank rev.#3 for pointing this out. The figure caption incorrectly stated analogue **6**, which has now been corrected to analogue **8**.
- 14. Supplementary figure S2. The images are rather small to confirm the morphological changes. Consider an alternative form of presentation.
 - We have altered the presentation of Supplementary figure S2 (now Supplementary Figure S3), in that an enlarged zone is now additionally presented.
- 15. Supplementary Figure S6. This should be presented as a Table. It is not clear why two *T. b. gambiense* genes appear in this list since the RNAi experiment involved *T. b. brucei*.
 - Supplementary Figure S6 has been transformed and labelled a Table.

Results from sequencing the RNAi inserts have been compared to genome database (BLAST search; TriTrypDB (<http://tritrypdb.org/tritrypdb/>)) and the results with the highest ranking match are depicted in the table, thereby giving rise to results originating from the *T. b. gambiense* genome.

- 16. Statistics. Number of animals used per group is not clear. Figure 4, P-values are not exact and the statistical method is not specified.
 - The number of animals per treatment group is now indicated in the methods section as well as under each figure displaying *in vivo* results.
The statistical method in Figure 4 is a Kruskal-Wallis test with Dunn's multiple comparison test.
P-values are generated by GraphPad PRISM and are presented in Figure 4 as the usual ranges (>0.05, >0.01 etc) and are not given as exact values.

Additional changes / additions made to the manuscript:

- Figure 3: initial figure did still contain original compound code (**FH7429_D**) in Panels B&C. these have been altered and now display **9**.
- In order to comply with journal guidelines, we have separated figure 5, which contained the adenosine transporter data, into a figure (inhibition plots) and a table (containing the Km & Ki values).
- Acknowledgements: the assistance of Dr. Jennifer Ann Black with the microscopy experiments were erroneously omitted, but has now been added. The sentence reads: "The authors express their gratitude to Dr. Jennifer Ann Black for assistance with the microscopy experiments." Some changes in the order of acknowledgment of the technical staff have been introduced according to the degree of contribution.

Reviewers' Comments:

Reviewer #2:

Remarks to the Author:

the authors have satisfied the concerns of my original review. accept the paper.

Reviewer #3:

Remarks to the Author:

I appreciate the authors' efforts to address the concerns raised by this reviewer. I find the revised manuscript to be more focussed and greatly improved through the additional experiments and corrections and would support publication as it stands.